# Contribution of Structure Learning Algorithms in Social Epidemiology: Application to Real-World Data

**DOI:** 10.3390/ijerph22030348

**Published:** 2025-02-27

**Authors:** Helene Colineaux, Benoit Lepage, Pierre Chauvin, Chloe Dimeglio, Cyrille Delpierre, Thomas Lefèvre

**Affiliations:** 1EQUITY Team, Centre d’Epidémiologie et de Recherche en Santé des POPulations (CERPOP), Institut National de la Santé et de la Recherche Médicale (INSERM)—Toulouse III University, 37 Allées Jules Guesde, 31062 Toulouse, France; 2Epidemiology Department, Toulouse Teaching Hospital, 37 Allées Jules Guesde, 31062 Toulouse, France; 3UMRS 1136, Pierre Louis Institute of Epidemiology and Public Health, Department of Social Epidemiology, Institut National de la Santé et de la Recherche Médicale (INSERM), Sorbonne University, 75005 Paris, France; pierre.chauvin@inserm.fr (P.C.); lefevre.thomas@gmail.com (T.L.); 4Toulouse Institute for Infectious and Inflammatory Diseases (INFINITY), Institut National de la Santé et de la Recherche Médicale (INSERM), UMR 1291, Centre National de la Recherche Scientifique (CNRS), UMR 5051, 31300 Toulouse, France

**Keywords:** causal discovery, directed acyclic graph, graphical models, social epidemiology, structure learning, Bayesian network, healthcare system utilization

## Abstract

Epidemiologists often handle large datasets with numerous variables and are currently seeing a growing wealth of techniques for data analysis, such as machine learning. Critical aspects involve addressing causality, often based on observational data, and dealing with the complex relationships between variables to uncover the overall structure of variable interactions, causal or not. Structure learning (SL) methods aim to automatically or semi-automatically reveal the structure of variables’ relationships. The objective of this study is to delineate some of the potential contributions and limitations of structure learning methods when applied to social epidemiology topics and the search for determinants of healthcare system access. We applied SL techniques to a real-world dataset, namely the 2010 wave of the SIRS cohort, which included a sample of 3006 adults from the Paris region, France. Healthcare utilization, encompassing both direct and indirect access to care, was the primary outcome. Candidate determinants included health status, demographic characteristics, and socio-cultural and economic positions. We present two approaches: a non-automated epidemiological method (an initial expert knowledge network and stepwise logistic regression models) and three SL techniques using various algorithms, with and without knowledge constraints. We compared the results based on the presence, direction, and strength of specific links within the produced network. Although the interdependencies and relative strengths identified by both approaches were similar, the SL algorithms detect fewer associations with the outcome than the non-automated method. Relationships between variables were sometimes incorrectly oriented when using a purely data-driven approach. SL algorithms can be valuable in exploratory stages, helping to generate new hypotheses or mining novel databases. However, results should be validated against prior knowledge and supplemented with additional confirmatory analyses.

## 1. Introduction

Science pursues three main types of results [1]: descriptive, predictive, or explanatory. Each objective entails specific methodological challenges and analytical approaches [2]. When the goal is explanatory (or etiological), the aim is to identify cause and effect relationships to act on the causes and alter the effects. The gold standard for causal inference is the randomized controlled trial. However, in epidemiology, randomized trials are not always feasible, requiring reliance on observational studies. Analyzing observational data, however, presents significant challenges for causal inference, as statistical associations derived from such data cannot be directly interpreted as causal effects. To address this, some authors have advocated for embracing the goal of causal inference while developing appropriate tools [3,4]. The structural causal model, developed by Judea Pearl [5], provides a robust methodological framework to achieve this goal. It integrates three key mathematical and philosophical tools: probability theory, which formulates causal relationships in probabilistic terms; counterfactual reasoning; and directed acyclic graphs (DAGs), a visual tool.

### 1.1. Epidemiology, Causality and Machine Learning

Machine learning (ML) methods, which are computational algorithms designed to optimize and automate modeling, were primarily developed for predictive purposes, making their application to causal inference, especially in epidemiology, less straightforward [6,7]. Causal inference typically relies on prior hypotheses and existing knowledge, which are not inherently derived from data [7,8,9,10]. While ML can enhance effect estimation by optimizing models, it does not inherently address the challenge of identifying confounders, a necessary step to assess causality when considering observational data [7]. Thus, ML methods are most beneficial for refining effect estimation using more flexible models than traditional regressions—such as VanderLaan’s targeted learning methodology [11,12]—once the causal structures have been established based on prior knowledge. However, these methods remain underutilized in practice: a 2021 review identified only eight studies applying ML to causal inference in social epidemiology [7].

Recently, a field known as ‘automatic discovery of causal structures using Bayesian networks’ (also referred to as ‘causal discovery’) has emerged [13]. These methods, developed as a distinct branch of machine learning [6], aim to automatically identify causal structures with varying levels of constraint from prior knowledge [9,13,14]. However, like other ML methods, they have not been widely adopted in epidemiological practice [13,14,15,16].

### 1.2. Machine Learning and Structure Learning: Exploring Potential Causality Thanks to the Structure of Data

Thus, a broader, specific branch of data analysis and ML in its widest acceptance has emerged to address the issue of the automatic or semi-automatic discovery of the underlying structure “hidden” in data which would account for complex variables relationships, may they be causal or not. These algorithms, referred to as structure learning algorithms, are of various types [17]. All these algorithms aim at delivering a representation of the data structure in terms of variables relationships as the output. This representation can be in the form of tables or vectors but usually are graphical outputs and presented as graphs in which each node stands for a variable and each edge between two nodes stands for a relationship between two variables. These graphs can be partially or totally directed graphs, meaning that there can be additional information besides nodes and edges, which is the direction of the relationship between the two variables. Mainly, graphs are constrained to be acyclic: there is no possibility for a node to be linked to itself through a more or less complex sequence of edges. At a minimum, SL algorithms deliver a graph without any orientations (totally unoriented). There are numerous references in the literature about so-called PDAG or DAG, applied to a wealth of scientific domains: partially directed or directed acyclic graphs.

More accurately, DAGs and PDAGs are specific outputs of SL algorithms, DAGs being defined as a graphical tool used in causal inference approaches [5], i.e., when the research objective is to estimate a causal effect from observational data. The principle is to visually represent all the variables of interest (the nodes) and all the possible causal links between these variables (oriented edges, i.e., arrows). The arrows are directed from a cause to an effect. Genealogical vocabulary is used to designate relationships between variables, such as parents, children, ascendants, and descendants. This tool makes it possible to be transparent about the assumptions concerning the underlying causal structure and to build the appropriate model to identify and estimate this effect, considering the context and thus avoiding important methodological biases such as not adjusting for a confounding factor or adjusting for a mediator [18]. DAGs are therefore the kind of SL algorithm output which is the closest to a causal representation of variable relationships, whereas the general SL algorithm output is focused on the overall data structure, with the variable relationships being causal or not. SL algorithms also provide quantified information, such as the strength of the association between two related variables. In epidemiological terms, the graphs produced alongside their quantified characteristics translate into networks of variables and determinants and the strength of associations.

### 1.3. Structure Learning and Social Epidemiology

Despite numerous ML and SL algorithms being developed using simulated data, challenges such as model parameterization, result validation, accuracy measurement, and generalizability hinder their application to real-world data [13]. Yet, these methods could be particularly valuable in clinical and epidemiological research, as they facilitate the exploration of complex phenomena in an intuitive manner, integrating both expert knowledge and empirical data [16]. In summary, SL methods could benefit epidemiology by (a) mining large datasets with numerous variables, enabling the simultaneous assessment of the roles of various variables and their direct or indirect contributions to a given outcome in a holistic and conditional framework; (b) serving as an alternative or complementary approach to traditional statistical inference, utilizing uncertainty quantification and propagation within a Bayesian framework rather than relying solely on frequentist methods; (c) enabling the translation of validated knowledge into actionable insights based on updatable data and observations.

However, the use of SL methods in social epidemiology still raises several questions. Confidence in the discovered structures, parameter estimates, and probability calculations remain uncertain, as the criteria for assessing reliability, robustness, and accuracy have not yet been established. Additionally, it is unclear whether the identified links are truly causal. These methods may not yet be mature enough for application to real epidemiological data, and the conditions under which they could be effectively used need to be clarified. So far, concerning biomedical topics, novel deep learning and non-deep learning structure learning algorithms have been tested solely on benchmark datasets [17]. Performances vary across the different tested algorithms, the main potential advantage of deep learning-based SL algorithms over other algorithms being their scalability. To the best of our knowledge, no study has tested SL algorithms on real-world datasets in social epidemiology against a non-automated approach based on experts’ knowledge.

The objective of this study was to delineate some of the potential contributions and limitations of structure learning methods when applied to social epidemiology topics and the search for determinants of healthcare system access. SL methods are seen as a proxy or a first step to approach the most complete and complex representation of causal relationships between variables, but causal discovery is not the main goal here. We focused on the overall structure of variable relationships. We tested three scenarios: (1) the structure of variable relationships as derived from two experts in social epidemiology and literature, tested against a real-life dataset (associations suggested by experts are “corrected” by the results of regression analyses performed on the data), (2) the structure of variable relationships as uncovered by three SL algorithms, and (3) the structure of variable relationships resulting from a mixed approach, i.e., the use of the same SL algorithms with some additional knowledge constraints. We compared the networks of interdependencies and assessed the presence, strength/stability, and finally the direction of links when provided by the algorithm. We therefore compared two opposite real-life approaches on a real-life dataset: one reflecting the usual habits of epidemiologists (reasoning based on experts and literature knowledge, then testing assumptions with statistical analyses on data) and another one being the use of an automated method to provide the whole data structure, which can then be challenged by experts. Our specific aim was to use these networks to identify the direct determinants of healthcare system utilization among various social candidate factors.

## 2. Materials and Methods

### 2.1. Population

The SIRS cohort (French acronym for Health, Inequality, and Social Disruptions) has been following a representative sample of approximately 3000 adults from the Paris metropolitan area since 2005 as part of a multidisciplinary research program designed to study the social and territorial determinants of health and healthcare utilization. In 2005, 50 neighborhoods (50 ‘IRIS’, i.e., census-based units each comprising around 2300 inhabitants and covering an average area of 0.25 km^2^) were randomly selected from the 2595 IRIS in Paris and the nearby departments of Hauts-de-Seine, Seine-Saint-Denis, and Val-de-Marne. Sixty households were then randomly drawn from each IRIS, and one adult per household was selected for interviews at their home. In 2010, 47% of the participants from the initial 2005 survey were successfully re-interviewed (2.6% had died, 1.8% were too ill to participate, 13.9% had moved out of the selected IRIS, 2.7% were unavailable during the survey period, 18.4% declined to participate, and 13.4% could not be contacted). Participants who could not be re-interviewed were replaced with newly selected individuals from the same IRIS to maintain representativeness. The refusal rate for new participants was 29%, consistent throughout both 2005 and 2010. This study utilizes data collected in 2010. The final sample of 3006 French-speaking adults was adjusted to account for the sampling strategy and then stratified by age and gender according to census data. There are no missing data in the dataset used. The detailed methodology of the SIRS study has been described previously [19,20,21].

### 2.2. Measures

The initial SIRS study focused on three key areas that justified the creation of the cohort: the impact of social ties and integration into various spheres of sociability on health-related behaviors, including the pursuit of curative and preventive care; the health status of immigrants and individuals of immigrant descent; and, finally, the influence of living environments, as captured by a geographic information system that integrates participants’ home addresses and some of their daily destinations [21]. The cohort is well known in the field of access-to-care studies in social epidemiology, having produced numerous publications on this topic [19,20,21,22,23,24,25].

In our study, healthcare utilization was the outcome. Since 2004, the French healthcare system has implemented a ‘soft gate-keeping’ model [26], allowing two types of ambulatory medical care access: (1) access to general practitioners (GPs) as a primary point of care or as an entry to specialists, and direct access to certain specialists (gynecologists, ophthalmologists, pediatricians, or psychiatrists); and (2) access to non-direct-access specialists, which typically requires a referral from a GP or, alternatively, direct consultation at full cost. Our primary outcome was the type of access, measured by the ‘Direct Access to Care’ (DAC) variable, coded as ‘yes’ if the individual had consulted a GP or a direct-access specialist at least once in the past twelve months. In a subsequent analysis, we explored the second type of healthcare utilization, measured by the ‘Indirect Access to Care’ (IAC) variable, coded as ‘yes’ if the individual had consulted a non-direct-access specialist at least once in the past twelve months.

The candidate determinants of healthcare system utilization were selected from the available data, the existing literature, and the experts’ knowledge encompassing variables related to health status, demographic characteristics, and socio-cultural and economic position. Health status was assessed using perceived health (categorized as ‘good’ or ‘average/poor’) and the presence of chronic health conditions. Demographic characteristics included age (grouped as ‘18–29’, ‘30–44’, ‘45–59’, ‘60–74’, and ‘75 or older’) and gender (women or men). Socio-cultural and economic position was measured using several variables: origin (categorized as French born to two French parents, French born to at least one foreign parent, and foreign-born immigrant), education level (none/primary, secondary, tertiary), employment status (employed, unemployed, inactive, or retired), income (total household income divided by the number of consumption units, sorted into quintiles), health insurance status (full coverage by statutory health insurance [SHI] and voluntary health insurance, or SHI coverage only), social integration (frequency of social contacts, categorized into quartiles), and proximity to the medical profession (having or not having a medical professional among close relatives). A detailed description of these variables has been provided previously [21,22,23].

The choice of these covariates results from two constraints: general experts’ knowledge in social epidemiology and what is available in the dataset. Therefore, the whole subsequent analyses, non-automated and automated, are solely based on the contents of the dataset and no other external considerations. As a result, not all relevant or potentially relevant variables associated with the outcomes can be analyzed in our study. This reflects the constraints of daily, real-life practice.

### 2.3. Statistical Analysis

We performed analyses based on the identification of a network of oriented links between the determinants of healthcare utilization, using several approaches. For the principal analysis, we used “direct access to care” as the outcome followed by “indirect access to care”. Considering the following network W→X →Y, where arrows denote oriented links between 3 variables, X is called a “direct determinant” (or “parent”) of Y (“child”), and W a “indirect determinant” (“grandparent”) of Y.

We used several approaches: first we used a non-automated approach and then structure learning approaches, based on several algorithms.

#### 2.3.1. “Non-Automated Approach”

An initial conceptual network was developed by two social epidemiology experts, drawing on the existing literature and prior knowledge. Each variable was then modeled by its potential direct determinants using logistic regression. A step-by-step selection process was applied to produce a final network, where all arrows represented significant associations (*p* < 0.05) between variables, with directions based on the initial expert-defined network. This final network, along with those produced by structure learning approaches, was visualized using the bnlearn and Rgraphviz packages in R. Links confirmed by the non-automated approach were represented with thick lines, while those proposed by experts but not confirmed were shown with thin lines.

#### 2.3.2. Automated (Structure Learning) Approaches

We constructed networks using several structure learning algorithms, which fall into three main categories [27,28]:Score-based algorithms: These identify the network that maximizes a score function reflecting how well the network fits the data [29]. We used the Hill Climbing algorithm with a BIC score (Bayesian Information Criteria) from this category.Constraint-based algorithms: These infer conditional dependencies between variables based on the Markov property of Bayesian networks and orient links using d-separation and acyclicity constraints, resulting in partially oriented networks [28,30,31]; we used the Inter-IAMB [32] algorithm.Pairwise algorithms: These employ an information-theoretic approach to filter out indirect interactions, resulting in non-oriented networks [33]. We used the ARACNE [33] algorithm.

Each algorithm was run 1000 times on bootstrap samples, and a link was included in the final network if its frequency in the bootstrap replicates was ≥5% (a conservative selection threshold). Initially, the algorithms were applied without constraints (“only data-driven learning”). We then introduced constraints (“constrained learning”) by specifying certain forbidden oriented links, i.e., ‘age’, ‘gender’, and ‘origin’ could not have any determinants, and ‘income’ could not be a determinant of ‘educational level’ or ‘employment status’. Additionally, ‘employment status’ could not influence ‘educational level’, and ‘health insurance status’ could not be a determinant of ‘income’, ‘educational level’, ‘employment status’, or having a medical professional among relatives.

#### 2.3.3. Comparison of Approaches

To streamline the results, we focus solely on the identified direct links with the outcome—specifically the connections between ‘access to care’ and other variables identified as direct determinants of ‘access to care’ or as dependent on it. For these links, we compare their presence or absence based on the different approaches used, their direction (whether they are directed toward or away from the outcome), and their strength. In the non-automated approach, strength is represented by the odds ratios in the final multivariate logistic regression. In the ‘structure learning’ approach, strength is measured by the frequency of the link’s selection in bootstrap replications (expressed as a percentage). While these measures of strength are not directly comparable, the ranking of link strength can still provide valuable insights.

Analyses were performed with R 4.3.1 [34] and R studio 1.2.5001. For the structure learning algorithms, we used the “bnlearn” package release 4.9 [28] and the Rgraphviz package release 2.46. Data on demand, codes, and output are available on GitHub “hcolineaux/SL_SocialEpi”.

### 2.4. Ethical Approval

The SIRS cohort received legal authorization from the “Comite consultatif sur le traitement de l’information en matière de recherche dans domaine de la sante” (CCTIRS) and from the “Commission nationale de l’informatique et des libertés” (CNIL): authorization no. 05-1024, 10 February 2005. Participants provided their verbal informed consent, as written consent was not required under French law [21,23].

## 3. Results

### 3.1. Description of Population

All 3006 individuals in the SIRS cohort were included in the analysis. Among the cohort, 60.5% were women, indicating a slight over-representation, and 89.2% of the population had accessed direct care at least once within the past year. Detailed population characteristics are provided in the Appendix A.

### 3.2. Non-Automated Epidemiological Approach

#### 3.2.1. Direct Access to Care

The final network, identified by the non-automated epidemiological approach from the conceptual network proposed by experts, is given in Figure 1. For direct links to DAC, the experts initially considered all candidate determinants as potential direct determinants of the outcome. Stepwise backward selection identified 5 of the 11 variables as direct determinants. The results are summarized in Table 1.

#### 3.2.2. Indirect Access to Care

Using a step-backward approach, 8 variables among the 11 candidate variables were confirmed as direct determinants: “age”, “gender”, “education level”, “income”, “health relatives”, “health insurance status”, “chronic disease”, and “perceived health status”. Detailed results are given in Appendix A.

### 3.3. Structure Learning Approaches

#### 3.3.1. Direct Access to Care

The data-driven Hill Climbing algorithms identified four variables directly linked to “direct access to care”: “gender”, “health insurance status”, “chronic disease”, and “perceived health status”. The data-driven ARACNE algorithms and Interleaved Incremental Association identified links between direct access to care for three variables: “gender”, “health insurance status”, and “chronic disease”. The relative strengths of the links seemed to follow the same order in each approach: the link with “gender” was always the strongest, followed by the presence of a chronic disease and “perceived health status”. “Age” was never retained in the structure learning approaches. The links were not always directed to the outcome. For example, the Hill Climbing and Interleaved Incremental Association algorithms identified “gender” as a child of “direct access to care”.

The knowledge constraints did not change the absence or presence of links, nor their relative strengths. However, they changed the direction of several links, particularly with the Hill Climbing models where the four links became oriented to the outcome; see also Figure 2. The final networks of all the approaches are given in Figure 3.

#### 3.3.2. Indirect Access to Care

The blind Hill Climbing algorithm identified the same eight direct determinants as non-automated approach. “Gender” and “education level” were no longer retained when knowledge-constraints were added. The Interleaved Incremental Association algorithms identified six of the eight variables (did not retain “income” and “health insurance status”) and selected one additional variable, “social integration”, which had the lowest strength. The ARACNE algorithm retained only “age” and “chronic disease”. The relative strengths of the links appeared to follow a consistent order across all approaches. The direction of the links was inconsistent. Applying constraints changed some directions. Further details are provided in the Appendix A.

## 4. Discussion

We compared a conceptual model, developed by experts based on prior knowledge and tested using stepwise logistic regression, to networks of interdependencies identified by various structure learning algorithms. The comparison focused primarily on the presence or absence of links, secondarily on their strength, and lastly on their direction, as our main objective was not about causal analysis but instead concerned uncovering the data structure. While the interdependency patterns and relative strengths are generally similar, the algorithms identify fewer links with the outcome than the non-automated approach. Additionally, the direction of some links between variables differs across methods. Introducing knowledge constraints results in networks that more closely resemble the non-automated approach.

Currently, most structure learning (SL) algorithms have been developed and validated using simulated data and are not widely applied to real-world data [13,14,15,16], which tend to be more complex, often contain missing values, and are typically smaller in size [35]. In this paper, we apply three of these algorithms to a real, complex, but well-known dataset to assess their ability to detect links that are well established in the literature. Our findings suggest that, despite their strong performance on simulated data, these methods are still challenging to implement on real-world datasets. This conclusion aligns with prior research comparing multiple algorithms for identifying causal factors of childhood diarrhea [35]. That study showed that the results are highly sensitive to the choice of algorithm, the handling of missing data, and the learning procedure, concluding that these methods are not yet mature enough to achieve reliable results. Over the past 30 years, many ‘causal discovery’ algorithms have been developed [9], primarily falling into two categories: score-based and constraint-based [36]. Kitson et al. provide a comprehensive overview of these algorithms and their evolution [37]. However, there is still a lack of simple, accessible guides to assist epidemiologists in selecting the appropriate algorithm.

### 4.1. About Uncovering the Data Structure Given Two Opposite Approaches

The choice of the ‘non-automated’ method is based on a frequent approach in epidemiology, though it is often implicit and highly debatable. It does not represent a gold standard per se but a real-life habit, and we do not consider its results as the undisputed “truth” in terms of the structure of the variable relationships. The goal here was to compare the consistency of results between a non-automated (what epidemiologists are taught to do facing observational data to analyze) and automated within an exploratory perspective method as a preliminary step before implementing a more in-depth causal analysis. Validating the results has been identified as a significant challenge when applying these methods to real data [13]. Specifically, it is difficult to validate results in real-world scenarios where the ‘true’ structure cannot be observed, making it impossible to measure the distance between the predicted and observed structures [13,38]. The results are often accepted if validated by experts, which may lead to circular reasoning and confirmation bias: the expert adjusts the results based on prior knowledge, and the outcome is interpreted as confirmation of this prior knowledge [13,39,40].

Regarding the automated methods, we chose a representative algorithm from each of the main families and use the bnlearn method. Other types of algorithms and packages, such as pcalg, could have been used. The aim was not to exhaustively evaluate all packages and algorithms but to test a few which are well-known with curated, open-source code on real data, in a context where recommendations that are accessible for epidemiologists are lacking. There are no clear guidelines on how to parameterize these algorithms either [37]. We therefore used standard parameters for all three algorithms and opted for a conservative threshold (links appearing in more than 5% of bootstrap samples) to minimize false negatives in this initial exploration of the data. However, all three algorithms appear less sensitive than the non-automated approach, which is consistent with the literature indicating that these methods require large samples to achieve robust results [9,36]. It may be necessary to adapt thresholds based on sample size and algorithm-specific power, making ‘human choices’ unavoidable. Unfortunately, no clear guidelines exist on this issue.

We used a very conservative threshold for selecting links between variables based on their frequency in bootstrap replications. Despite this, and although the interdependence networks and relative strengths are similar, the algorithms identify fewer links with the outcome compared to the non-automated approach. Relationships between variables are sometimes considered misdirected in the purely data-driven approach, whereas the non-automated model appeared more intuitively accurate (e.g., gender → DAC). Adding knowledge constraints adjusts some link directions, making the networks more consistent with the non-automated model when ‘direct access to care’ is the outcome. This is also true for ‘indirect access to care’ when using the Interleaved Incremental Association algorithm. Additionally, none of the algorithms identify a link between age and the outcome, possibly due to the multinomial nature of the age variable. Even though recent algorithmic developments can handle various types of variables, the effectiveness may still depend on the variable’s form. Binary or continuous Gaussian variables are generally easier for these algorithms to process.

### 4.2. A Word About Causal Interpretation of Variables Networks

Although our main objective was not to explore causal networks, but rather to explore the potential interests of structure learning algorithms to uncover or help uncover the data structure of a real-life dataset, the variety of SL algorithms implies that they provide different kinds of outputs. Some algorithms (ARCANE, for example) deliver only unoriented graphs, while other ones deliver partially or fully DAGs. Therefore, some algorithms suggest that there are oriented links between variables, implying a certain conception of causality between these two variables. We provide commentary on these aspects here.

Several assumptions are commonly made when interpreting learned networks as causal networks: (1) the Causal Markov condition, which states that all variables are independent of their non-descendants, conditional on their parents (direct causes) [41]; (2) Faithfulness, which posits that causally connected variables are probabilistically dependent—this assumption may fail if the effects of multiple paths cancel each other out, rendering the cause and effect probabilistically independent [42]; and (3) Causal Sufficiency, which assumes that there are no unobserved common causes (confounders) [43]. These assumptions are nearly impossible to satisfy with real data, but the same is true for many assumptions underlying non-automated statistical methods, such as normality. This may explain the misoriented links observed in our results. Incorporating expert knowledge is essential, at least to exclude impossible directed links (e.g., income → sex) and to correct potential misorientations [35,44]. Selection biases are also challenging to identify and account for, whether the approach is automated or not. For example, an observed link between ‘age’ and ‘origin’ likely results from selection bias (collider bias), but the consequences on the results are complex to assess for both approaches.

### 4.3. Are Structure Learning Algorithms Fit for (Social) Epidemiologists’ Use?

Despite these limitations, structure learning holds significant promise for epidemiology. First, these methods are more interpretable than other machine learning models because they use visual representations [35]. The increasing complexity of machine learning models has amplified the ‘black box’ issue, making them difficult to use, evaluate, and interpret, especially for clinical decision-making [45]. In contrast, structure learning relies on graphical (usually) Bayesian networks, which can be considered a more ‘explainable’ machine learning method, at least in terms of result interpretation [46]. Second, these methods are specifically designed for identifying potential causal structures based on the identification of data structure rather than purely predictive goals, which is particularly relevant for epidemiology, where causal inference is central. However, epidemiologists must be fully aware of the type of causality being addressed: the causality considered by data scientists often differs from that sought by epidemiologists. Unless specific assumptions—such as no hidden confounders and normality for all continuous variables—are met, the directed link established by structure learning algorithms is only of the informational type. This means that the direction of a link is determined by how one variable informs about another. For example, age may provide more information about socio-economic status rather than vice versa. For an epidemiologist, a causal link between from X to Y refers to the fact that an intervention in X changes Y [4]. Therefore, the chosen direction between two characteristics might differ from that derived from data and related conditional probabilities. Once this distinction is understood, we can use the term ‘causality’ with greater confidence.

The use of SL algorithms to uncover the data structure based on real-life data may be of interest, as they may be more documented in terms of settings (parameter tuning and sensitivity, to name a few). Their further use as tool for direct uncovering causality relationships is out of the scope of our study, but the current limitations in terms of structure identification suggest that epidemiologists should be very careful when considering this option. It may be more prudent to use them initially as complementary tools for uncovering and testing data structure, and, based on these results, to test causal hypothesis.

## 5. Conclusions

Our study highlights two important issues. The first pertains to disciplinary differences, not only in terms and concepts but also in applications and objectives. Both epidemiologists and data scientists work with data, but their approaches differ significantly. Epidemiologists rely on statistical methods to compensate for deviations from an experimental and controlled design, reasoning primarily in terms of experiential evidence. Data scientists, on the other hand, often adopt a data-driven approach, treating data as the reality of the field itself, sometimes at the expense of considering the actual context. This can lead to a confusion of territory, where conclusions drawn from data may hold true for one context but not for the other. Consequently, the meaning of directed link diverges: epidemiologists seek to identify mechanisms linking exposure to disease, while data scientists focus on the best representation of the data in an informational perspective but are maybe too often prone to confuse informational and mechanistic causality. The second issue relates to the potential of structure learning algorithms. Our study shows that the current diversity of these algorithms, the lack of clear guidelines for parameter selection, and the implications of these choices prevent us from recommending their use without caution. Epidemiologists lacking a solid theoretical background or technical support in this area may find it challenging to use these methods safely and effectively. Given the rapid advancement of these techniques and their adoption across disciplines, it is crucial for epidemiologists to engage with them. This engagement is necessary not only to identify appropriate applications but also to determine what conclusions can be reliably drawn from these methods.

## Figures and Tables

**Figure 1 ijerph-22-00348-f001:**
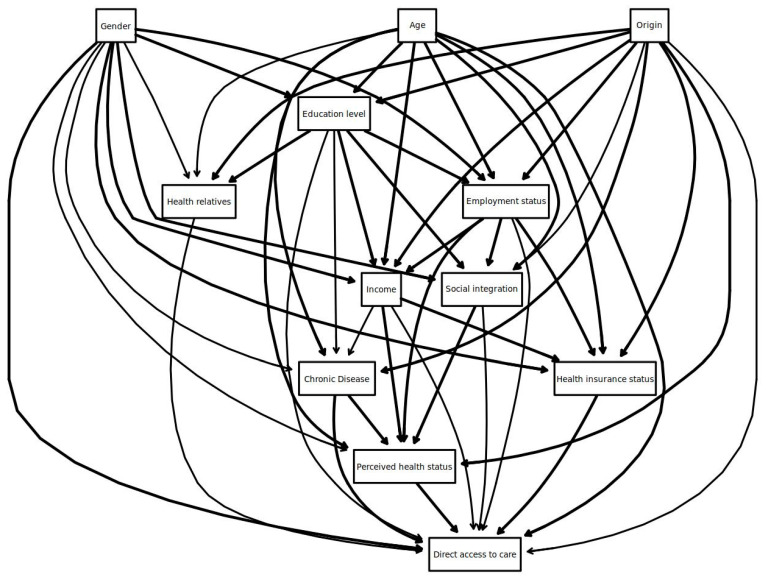
The network identified by a non-automated epidemiological approach. The solid thick lines indicate links proposed by experts and confirmed by the non-automated epidemiological approach, thin lines indicate links considered as potential by experts but not confirmed by the approach. The direction of the arrows was defined by an expert (not data driven). The graph was produced by bnlearn release 4.9 and Rgraphiz package in R.

**Figure 2 ijerph-22-00348-f002:**
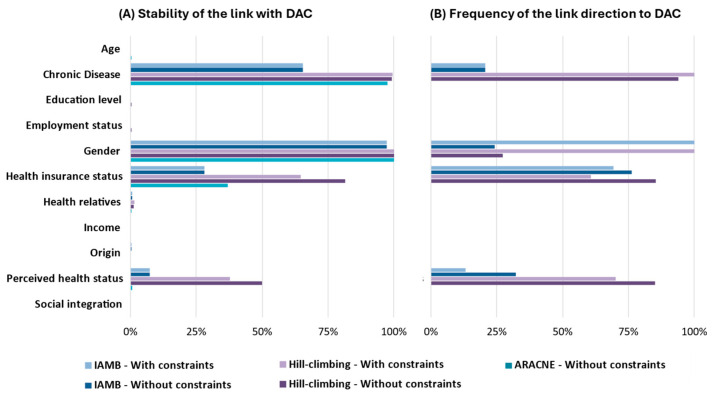
Results of structure learning approach with “direct access to care” (DAC) as outcome. (**A**) Strength/stability is the relative apparition of the link among the bootstrap replicates; (**B**) frequency of the direction of the link to the DAC (direct access to care) variable among the bootstrap replicates; on the graphs, an arrow points to the DAC if the frequency is ≥50%, and points from the DAC otherwise. IAMB = Interleaved Incremental Association.

**Figure 3 ijerph-22-00348-f003:**
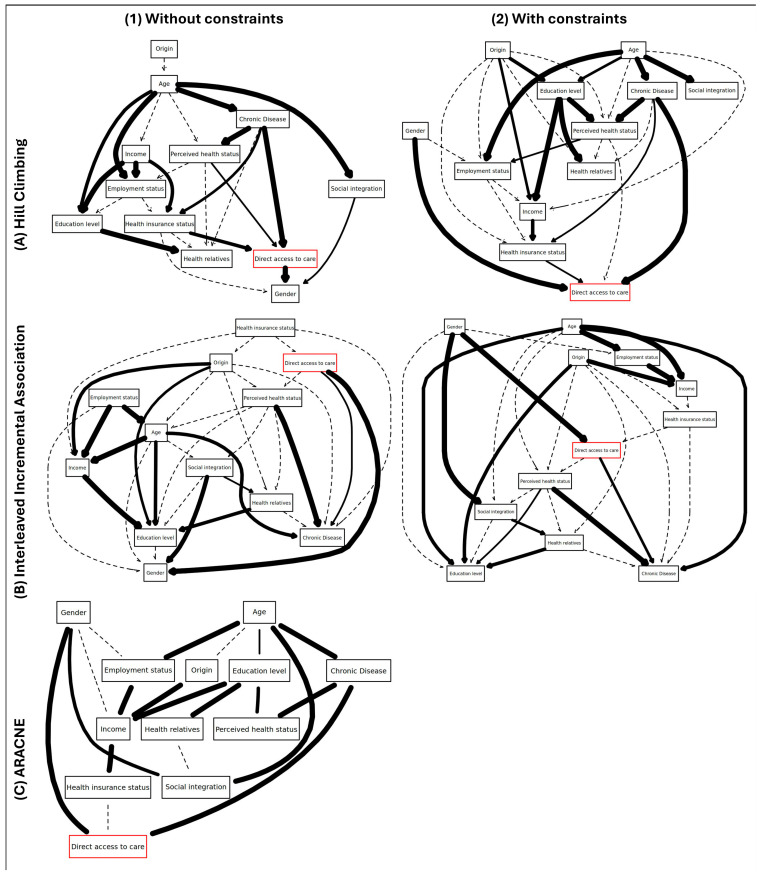
Final networks produced without (**1**) and with (**2**) knowledge constraints with Hill Climbing (**A**), Interleaved Incremental Association (**B**), and ARACNE (**C**) algorithms. The lines are plotted if the relative frequency of the links is ≥0.05 in the bootstrap replicates (*n* = 1000). When the relative frequency of the links in the bootstrap replicates is lower than 0.5, the lines are plotted as dashed lines. When the relative frequency of the links in the bootstrap replicates are higher than 0.5, the thickness of the line is proportional to the relative frequency.

**Table 1 ijerph-22-00348-t001:** Results of non-automated approach with “direct access to care” (DAC) as outcome.

		Link to DAC ^1^	Direction ^2^	OR, 95%CI ^3^
Age	18–29	Yes	To DAC	Ref
	30–44			1.39 (0.98 to 1.98)
	45–59			1.20 (0.83 to 1.72)
	60–74			1.72 (1.12 to 2.64)
	75+			2.23 (1.22 to 4.31)
Gender	Men	Yes	To DAC	Ref
	Women			3.13 (2.45 to 4.02)
Origin	3 categories	No	-	-
Education level	3 categories	No	-	
Employment status	3 categories	No	-	
Income	5 categories	No	-	
Health insurance status	None or SHI only	Yes	To DAC	Ref
	SHI and VHI			2.38 (1.75 to 3.23)
Health relatives	2 categories	No	-	
Social integration	4 categories	No	-	
Chronic disease	No	Yes	To DAC	Ref
	Yes			2.94 (2.12 to 4.16)
Perceived health status	Good	Yes	To DAC	Ref
	Bad–Average			2.16 (1.50 to 3.20)

^1^ “yes” if significant association found in final logistic regression modeling DAC by other variables identified as its potential direct determinants by experts; ^2^ based on the initial network defined by expert; ^3^ OR, 95%CI are the Odds Ratio and its 95% confidence interval as estimated by the final logistic regression.

## Data Availability

The authors confirm that all data underlying the findings are fully available without restriction. All relevant data have been deposited to Dryad (DOI:10.5061/dryad.9v79s) cf article Plos One healthcare utilization. Requests can be made to the ERES team (UMRS 1136, Pierre Louis Institute of Epidemiology and Public Health, Department of Social Epidemiology, INSERM, Sorbonne University, Paris, France) for access to the data.

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
