# Peer review of "Contribution of Structure Learning Algorithms in Social Epidemiology: Application to Real-World Data"

_ijerph, 2025, doi:10.3390/ijerph22030348_

Round 1
Reviewer 1 Report (Previous Reviewer 3)
Comments and Suggestions for Authors
The responses are acceptable.
Author Response
Thank you for your feedback
Reviewer 2 Report (Previous Reviewer 2)
Comments and Suggestions for Authors
1. My queries are answered promptly and responsibly.
2. I recommend just minor revisions on the English language grammar and clarity.
Author Response
Thank you for your feedback. We made some corrections for clarity.
Reviewer 3 Report (Previous Reviewer 1)
Comments and Suggestions for Authors
The paper provides a comprehensive and well-structured exploration of the role of structure learning algorithms in social epidemiology, offering valuable insights into their potential applications and limitations. Its balanced discussion, combining expert-driven and automated approaches, makes a significant contribution to advancing the integration of machine learning techniques in epidemiological research.
Author Response
Thank you for your feedback
This manuscript is a resubmission of an earlier submission. The following is a list of the peer review reports and author responses from that submission.
Round 1
Reviewer 1 Report
Comments and Suggestions for Authors
- You should also use the present tense in certain parts of the abstract, such as when stating the objectives or the general facts of the study, or when presenting conclusions and interpretations.
- Many of the references cited are not recent, which may undermine the paper's relevance in the rapidly evolving field of machine learning and causal inference.
- While the authors apply multiple structure-learning algorithms, the sensitivity of these methods to uncontrolled variables is not thoroughly discussed. Addressing how missing controls impact the algorithms' performance is necessary.
- Major revisions are needed in methodology, especially by incorporating experimental controls or rigorous statistical adjustments to handle missing variables.
-Use a bar chart to compare the frequency or strength of identified links across different structure-learning algorithms. Each bar could represent a variable (e.g., age, chronic disease), and separate bars could show the results of each algorithm, such as the non-automated approach vs. structure learning.
Are there any patents for this study? If yes, please include the patent number or application reference. Otherwise, there is no need for a section titled 'Patents'.
Comments on the Quality of English LanguageThe paper frequently uses past tense in sections where present tense would be more suitable, especially when discussing findings, general facts, and the implications of the results. Using the present tense for these parts would improve readability and emphasize the ongoing relevance of the study.
Author Response
Reviewer 1
1 - You should also use the present tense in certain parts of the abstract, such as when stating the objectives or the general facts of the study, or when presenting conclusions and interpretations.
We made modifications to the abstract accordingly.
2 - Many of the references cited are not recent, which may undermine the paper's relevance in the rapidly evolving field of machine learning and causal inference.
We agree with the reviewer that machine learning is evolving rapidly, more specifically, there is a wealth of publications falling into the machine learning domain, which does not mean significant new results or game changing techniques appear every day. It is basically impossible to test a representative set of machine learning algorithms on a given dataset, and it is also irrelevant. Besides, the fact is that structure learning principles and algorithms are not growing so fast, and more important, haven’t been applied to social epidemiology.
Therefore, our choice was to focus on well-known and validated SL algorithms, with curated and available code before all other considerations. Maybe there are some algorithms which would perform better on our data: here, we are not pursuing percentages of accuracy, but the assessment of the potential overall relevance of SL in social epidemiology.
We added some references regarding causal inference, DAGs and SL learning applied to health, especially Upadhyaya P et al. Scalable Causal Structure Learning: Scoping Review of Traditional and Machine Learning-Based Methods With Applications in Biomedicine. JMIR Med Inform. 2023;11:e42557.
3 - While the authors apply multiple structure-learning algorithms, the sensitivity of these methods to uncontrolled variables is not thoroughly discussed. Addressing how missing controls impact the algorithms' performance is necessary.
The reviewer raises an important point. We have the feeling we were not clear enough about the main objective of the study. We talked about causal inference and causal structures since it is somehow one of the « ultimate » goals of epidemiology. But the main goal of the study is not to assess directly the potential discovery of causal structure, only of data structure in the sense of the network of complex relationships or associations between variables, may they be causal or not.
We tried to reframe and rephrase these aspects to make it clearer (see for example the revised abstract and the widely revised introduction).
From that perspective, while we agree with the reviewer about the point they raise, it is in our study not the main limitation to be discussed. Besides, more than missing controls, as with any other algorithms or analysis techniques, it is obvious that if there is any « hidden » variable in the system we are trying to analyze, the resulting structure won’t be the most representative or even correct considering « reality ».
Nonetheless, the reviewer can find in the discussion the following comment about the causal interpretation of the learnt structure:
“Several assumptions are commonly made when interpreting learned networks as causal networks: […] (3) Causal sufficiency, which assumes no unobserved common causes (confounders) [36]. These assumptions are nearly impossible to satisfy with real data, but the same is true for many assumptions underlying non-automated statistical methods, such as normality. This may explain the misoriented links in our results.”
Finally, our perspective is to compare, given a real-world observational dataset and two opposite approaches (expert’s knowledge driven vs algorithm data driven), the results of these two approaches where no “gold-standard” is available. As with any observational dataset, the presence of “missing” or “hidden” variables is to be expected. However, we are not able to correctly assess their impact on the relative performances of each approach because those hidden variables are unknown to us.
4 - Major revisions are needed in methodology, especially by incorporating experimental controls or rigorous statistical adjustments to handle missing variables.
As stated above in 3, we agree with the reviewer and would have proceeded otherwise if our goal had been to only and primarily interpret learnt structures as pure causal structures, which is not the case. The expert’s knowledge driven approach is prone to similar biases (confounding bias, but also selection or measurement bias) for which numerous sensitivity approaches have been described in the literature. However, the issues related to these biases were beyond the objectives of our analysis. The point is that our starting point was to use a well-known and well-built real-world cohort in social epidemiology as well as the expertise of social epidemiologists to assess if SL algorithms 1) could retrieve a raw data structure which is meaningful and if possible, accurate 2) compared to what we know in the domain from previous analyses of the same cohort.
So, the main objective was not to build the most complete and accurate causal structure of healthcare system utilization, but to explore the potential of SL techniques on a specific dataset. Obviously, it is always likely that some variables may fail to account for the whole process.
Additionally, if it had been our objective, we are very curious to know what kind of experimental controls the reviewer could think of: if observational studies exist, it is mostly because experimental designs are out of reach for many topics. It is all the truer in the case of social epidemiology.
5 - Use a bar chart to compare the frequency or strength of identified links across different structure-learning algorithms. Each bar could represent a variable (e.g., age, chronic disease), and separate bars could show the results of each algorithm, such as the non-automated approach vs. structure learning.
We have replaced table 2 with figure 2 in order to present the main results with a bar chart as proposed.
Are there any patents for this study? If yes, please include the patent number or application reference. Otherwise, there is no need for a section titled 'Patents'.
No, there is no patent here. We only used a well-known cohort data and open-source curated code, available on GitHub “hcolineaux/SL_SocialEpi” (access added at the end of the Method section).
6 - Comments on the Quality of English Language: The paper frequently uses past tense in sections where present tense would be more suitable, especially when discussing findings, general facts, and the implications of the results. Using the present tense for these parts would improve readability and emphasize the ongoing relevance of the study.
We made changes according to the reviewer’s remark.
Reviewer 2 Report
Comments and Suggestions for Authors
These are my professional comments and suggestions about the paper:
FOR THE INTRODUCTION:
1. The motivation of the study for me is ambiguous, specifically about the limitations of current epidemiological approaches or methods in a causal inference. I would like the authors to explicitly detail how structure learning may address the paper's limitation, this would clarify the unique contribution of the research.
2. In my professional opinion, the use of terms like "causal discovery" and "directed acyclic graphs" are mentioned without enough explanation. Please clarify the concepts early as this would clear things up for the broader audience specifically to readers who are not familiar with statistical methods.
3. I would like the proponents to discuss existing critiques of structure learning methods to frame the study's unique approach. By addressing these critiques, it would show a deep understanding of the field and position the study as a response to existing limitations.
4. The paper lacks a theoretical framework, this leaves the study aim to open interpretation. Please establish a guiding theoretical basis for the use of structure learning methods. This will surely clarify the underpinnings and provide a reference point for interpreting results.
FOR THE REVIEW OF RELATED LITERATURE:
5. For me the literature review narrowly concentrated on structure learning and lacks the references to foundational work in causal interference. Please find seminal papers on causal inference, such as those work by Pearl and Rubin.
6. Also I would like to point out that there is very little discussion of alternative methods for causal discovery. It would not hurt to compare structure learning with other causal inference methods such as propensity score matching or instrumental variable analysis. This will also justify your choice of structure learning algorithms.
7. How about recent advancements in machine learning? Please include recent development in this field such as deep learning approaches to causal inference. This will convey awareness of cutting-edge techniques.
8. I would like the author to include about discussions of interdisciplinary perspectives on structure learning from various domain such as statistics, artificial intelligence or network science. Please highlight the insights from these disciplines to increase the study's theoretical foundation.
9. If you are doing the literature review, it is important to include, if there is any, contradictory findings in the literature about structure learning. It is indeed important to address conflicting results to have a balanced view and show deeper engagement with the field. This will sure increase the credibility of the review.
FOR THE METHODS
10. For my professional experience the SIRS cohort selection lacks detail about randomization and control for confounding factors. Your initial sampling strategy is briefly outlines, however, there is no indication of how the final sample of 3,006 was refined or adjusted for bias from attrition. Also, I find it lack transparency on whether weighting adjustments were applied. Please address the potential sampling biases, this is critical for the validity and generalizability of the study.
11. I have also observed that the paper has list of determinants for healthcare access but does not show a clear rationale for each. Explaining why variables such as "social integration" and "proximity to medical professionals" were chosen over other potential determinants would provide more insight into the study’s theoretical foundation. In addition, it is unclear whether these variables were selected based on existing empirical evidence or theoretical relevance. Please detail the criteria for variable inclusion and exclusion. This would improve transparency and replicability. Without a justified selection criterion, the validity of results could be questioned based on my experience.
12. Please include validation methods such as cross-validation or comparison with an established causal model for the evaluation of each structure learning. By ensuring the evaluation metrics used and the validation in place. It is important for confidence in the algorithms' outputs
13. I suggest to include a sensitivity analysis for the assessment of the model's robustness. Given the complexity of causal inference in observational data. The sensitivity analysis is important. This omission weakens the reliability of the causal conclusions drawn from the data.
FOR THE RESULTS:
14. There are results about network structures generated by different algorithms but it does not offer sufficient interpretative detail. For me it is unclear how readers should interpret the directionality or thickness of the links in these structures. Detailed legends as well as interpretative commentary on the link strengths can aid in clarity. Provide also comparative visualizations of network outputs across algorithms to improve network structure interpretability.
15. The results section implies causality in some findings without sufficient backing. Causal claims should be carefully qualified, especially in observational studies, where inferring causality is challenging. Highlighting that the findings are associative rather than strictly causal would be a more accurate interpretation.
Comments on the Quality of English Language
1. Moderate English revisions needed.
Author Response
FOR THE INTRODUCTION:
- The motivation of the study for me is ambiguous, specifically about the limitations of current epidemiological approaches or methods in a causal inference. I would like the authors to explicitly detail how structure learning may address the paper's limitation, this would clarify the unique contribution of the research.
We thank the reviewer for their remark. Throughout all reviewers’ remarks, we had the feeling we failed in stating correctly the main objective of our study.
The main objective of our study was not directly related to causal inference. We mentioned causal inference since it is one if not the main ultimate goal in epidemiology: to search for causes of health events, for example. On the other hand, epidemiologists often deal with potentially multivariable (massively multivariable) topics, for which the relationships and interactions are hard to account for simultaneously and accurately. Their main material is observational data, and not experimental data with controls, so that approaching causality is all the more a difficult task.
Something intermediary would be to have efficient tools able to account of multivariable relationships in observational data, so that it would provide an overall picture of a given topic. Based on such a synthesis of variables interactions, causality (multicausality) could be explored not « blindly » but with some kind of imperfect map of the territory which could be improved with time.
We tried to reframe and rephrase the context and the objective of our study. We made important changes in the abstract and widely rearranged the whole introduction.
- In my professional opinion, the use of terms like "causal discovery" and "directed acyclic graphs" are mentioned without enough explanation. Please clarify the concepts early as this would clear things up for the broader audience specifically to readers who are not familiar with statistical methods.
As previously said, these concepts are not central in the study but should be considered as elements of context and far goals to reach. We added information and references about causal discovery and DAGs and made clearer the relationships between causal discovery and structure learning. For example:
“Directed Acyclic Graphs (DAGs) are a graphical tool used in causal inference approaches (Pearl et Mackenzie, 2018), i.e. when the research objective is to estimate a causal effect from observational data. The principle is to visually represent all the variables of interest (the nodes) and all the possible causal links between these variables (the arrows). The arrows are directed from a cause to an effect. Genealogical vocabulary is used to designate relationships between variables, such as parents, children, ascendants and descendants. This tool makes it possible to be transparent about the assumptions concerning the underlying causal structure and to build the appropriate model to identify and estimate this effect, taking into account the context and thus avoiding important methodological biases such as not adjusting for a confounding factor or adjusting for a mediator, etc. (Tennant et al., 2021).”
Pearl, Judea et Dana Mackenzie (mai 2018). The Book of Why : The New Science of Cause and Effect. 1er edition. Penguin.
Tennant, Peter WG, Eleanor JMurray, Kellyn F Arnold et al. (2021). ≪ Use of directed acyclic graphs (DAGs) to identify confounders in applied health research : review and recommendations ≫. In : International journal of epidemiology 50.2, p. 620-632.
- I would like the proponents to discuss existing critiques of structure learning methods to frame the study's unique approach. By addressing these critiques, it would show a deep understanding of the field and position the study as a response to existing limitations.
In the discussion section, we discussed the fact that there are some criticisms against SL methods, such as their high sensitivity to the chosen SL algorithms. See for example:
“Our findings suggest that, despite their strong performance on simulated data, these methods are still challenging to implement on real-world datasets. This conclusion aligns with similar research comparing multiple algorithms for identifying causal factors of childhood diarrhea [28]. That study showed that results were highly sensitive to the choice of algorithm, handling of missing data, and learning procedure, concluding that these methods are not yet mature enough to achieve reliable results.”
Our study is not designed to address these issues on a theoretical basis. It is only designed to explore the potential of classical SL algorithms in discovering a meaningful and if possible, the most accurate data structure on real world data.
- The paper lacks a theoretical framework, this leaves the study aim to open interpretation. Please establish a guiding theoretical basis for the use of structure learning methods. This will surely clarify the underpinnings and provide a reference point for interpreting results.
Our intention here is not to provide a course or a primer on DAGs, causal inference or structure learning. There exists a few. Our purpose is to test two opposite approaches when applied to the same real-life dataset, and reflecting real-life, practical habits.
We added some information about theoretical framing of epidemiology and causal search, some details about DAGs and graphs, as well as more details between structure learning and causal discovery.
FOR THE REVIEW OF RELATED LITERATURE:
- For me the literature review narrowly concentrated on structure learning and lacks the references to foundational work in causal interference. Please find seminal papers on causal inference, such as those work by Pearl and Rubin.
As stated previously, causal inference is of secondary importance in our study, not the main objective. We added some information about some more theoretical perspectives, such as the following:
“Science can pursue three main objectives (Hernán, Hsu, and Healy, 2019): descriptive, predictive, or explanatory. Each objective entails specific methodological challenges and analytical approaches (Shmueli, 2010). When the goal is explanatory (or etiological), the aim is to identify cause-and-effect relationships to act on the causes and alter the effects. The gold standard for causal inference is the randomized controlled trial.
However, in epidemiology, randomized trials are not always feasible, requiring reliance on observational studies. Analyzing observational data, however, presents significant challenges for causal inference, as statistical associations derived from such data cannot be directly interpreted as causal effects. To address this, some authors have advocated for embracing the goal of causal inference while developing appropriate tools (Hernán, 2018; Pearl, 2009). The structural causal model, developed by Judea Pearl (Pearl and Mackenzie, 2018), provides a robust methodological framework to achieve this goal. It integrates three key mathematical and philosophical tools: probability theory, which formulates causal relationships in probabilistic terms; counterfactual reasoning; and directed acyclic graphs (DAGs), a visual tool.”
Ref :
Hernán, Miguel A, John Hsu et Brian Healy (2019). « A second chance to get causal inference right : a classification of data science tasks ». In : Chance 32.1, p. 42-49.
Shmueli, Galit (2010). « To explain or to predict ? » In : Statistical science 25.3, p. 289-310
Hernán, Miguel A (2018). « The C-word : Scientific euphemisms do not improve causal inference from observational data ». In : American journal of public health 108.5, p. 616-619.
Pearl, Judea (sept. 2009). Causality. Google-Books-ID : LLkhAwAAQBAJ. Cambridge University Press. isbn : 978-1-139-64398-6.
Pearl, Judea et Dana Mackenzie (mai 2018). The Book of Why : The New Science of Cause and Effect. 1er edition. Penguin.
- Also I would like to point out that there is very little discussion of alternative methods for causal discovery. It would not hurt to compare structure learning with other causal inference methods such as propensity score matching or instrumental variable analysis. This will also justify your choice of structure learning algorithms.
We understand the point of the reviewer and would agree if our main objective had been causal discovery or inference. It was not. It was about assessing and comparing the obtained data structure in terms of networks of variables associations (causal or not) with experts’ knowledge. From this perspective, and to the best of our knowledge, there is no point and no other techniques to compare here. Propensity scores and instrumental variables cannot help to uncover causal structures. These methods are used to estimate specified causal estimands, by trying to approach a quasi-experimental setting with observational data. They rely on structural assumptions that are assumed to be correct in order to assess the identifiability of the estimand and then to obtain an unbiased estimate. The need to specify a causal structure (by expert’s knowledge or using structure learning algorithms) is a task that necessarily lies upstream of the estimation of a causal estimand between an exposure and an outcome of interest. .
- How about recent advancements in machine learning? Please include recent development in this field such as deep learning approaches to causal inference. This will convey awareness of cutting-edge techniques.
As said, it is not about causal inference but structure learning. Nonetheless, we added references, especially this one: Upadhyaya P et al. Scalable Causal Structure Learning: Scoping Review of Traditional and Machine Learning-Based Methods With Applications in Biomedicine. JMIR Med Inform. 2023;11:e42557.
The study is recent (2023) and the authors explored several structure learning algorithms on several benchmark datasets, including deeplearning-based SL algorithms.
So far, these techniques still need validation and development, and their performances are not obviously better than non-deeplearing SL algorithms. Their most promising advantage would be scalability. We added information about this in the abstract, introduction and discussion.
- I would like the author to include about discussions of interdisciplinary perspectives on structure learning from various domain such as statistics, artificial intelligence or network science. Please highlight the insights from these disciplines to increase the study's theoretical foundation.
We understand the point of the reviewer, but feel that it is not in line with our objectives and will tend to make our points even more confuse. Our main perspective is the epidemiologist’s perspective, in real-life settings. There are papers about technical and theoretical aspects of algorithms. We added some references considering this. We also already discussed the fundamental difference of point of view between disciplines such as epidemiology and data science.
- If you are doing the literature review, it is important to include, if there is any, contradictory findings in the literature about structure learning. It is indeed important to address conflicting results to have a balanced view and show deeper engagement with the field. This will sure increase the credibility of the review.
We are not sure to understand the reviewer’s point here. We did not do a literature review here – it would be another kind of study. We do feel we are critical regarding SL approaches. We do not understand what « contradictory findings about structure learning » mean precisely. If it is about the fact that different SL algorithms can lead to different results when applied on the same data, for example, we stated it, such as in:
“Our findings suggest that, despite their strong performance on simulated data, these methods are still challenging to implement on real-world datasets. This conclusion aligns with similar research comparing multiple algorithms for identifying causal factors of childhood diarrhea [28]. That study showed that results were highly sensitive to the choice of algorithm, handling of missing data, and learning procedure, concluding that these methods are not yet mature enough to achieve reliable results.”
FOR THE METHODS
10. For my professional experience the SIRS cohort selection lacks detail about randomization and control for confounding factors. Your initial sampling strategy is briefly outlines, however, there is no indication of how the final sample of 3,006 was refined or adjusted for bias from attrition. Also, I find it lack transparency on whether weighting adjustments were applied. Please address the potential sampling biases, this is critical for the validity and generalizability of the study.
The SIRS (Santé, Inégalités et Ruptures Sociales) is a well-known cohort in social epidemiology with a wealth of scientific publications based on its data. Our sample is not a sub-sample of the cohort, it is the whole cohort population. Social determinants are difficult to study since they are related to people and populations usually underrepresented in studies or hard to reach for several reasons. The SIRS cohort has been designed with these issues in mind and it has been built based on a three level, representative sampling strategy (neighbourhood, household, individual) with an over representation of deprived areas. Weighting and scaling applied a posteriori but of minor impact compared to raw data. The cohort has been described in different articles or book chapters. As for any known and published cohorts, we do not report in our article the whole construction of the cohort population. See for example Chauvin P, Parizot I. Les Inégalités Sociales et Territoriales de Santé Dans L’agglomération Parisienne. Une Analyse de La Cohorte Sirs (2005) Saint-Denis La Plaine: Délégation interministérielle à la ville; 2009. pp. 1–105.
Through our use of SL algorithms, we did not use specific weighting – most algorithms are not made to take into account weights.
Besides, the remark is particularly sound in the context of the search for causal relationships. Again, this is not the goal here, nor to account for all possible determinants in social epidemiology. Given these data, and given what is known both in social epidemiology and published based on this cohort, are SL algorithms able to retrieve the same structure of variables relationships than experts and literature? That’s the question here. The role of missing confounders or of the sampling strategy leading to the constitution of the cohort is not relevant for our objective here.
- I have also observed that the paper has list of determinants for healthcare access but does not show a clear rationale for each. Explaining why variables such as "social integration" and "proximity to medical professionals" were chosen over other potential determinants would provide more insight into the study’s theoretical foundation. In addition, it is unclear whether these variables were selected based on existing empirical evidence or theoretical relevance. Please detail the criteria for variable inclusion and exclusion. This would improve transparency and replicability. Without a justified selection criterion, the validity of results could be questioned based on my experience.
We understand the reviewer’s remark. Nonetheless, doing what would be ideal and asked here is mainly unrealistic. It would not take several weeks and a handful of skilled researchers, but months if not years and several research teams to achieve perfectly that task. That’s why in any similar settings, we require the insight of experts in the domain. Their opinion acts as a gold standard where there is no comprehensive synthesis available in the domain. That’s all the point of the potential use of SL algorithms: to help researchers gather and synthetize complex and numerous variables relationships in topics which are not purely mechanistics or deterministics with only one or two causal factors. So, the choice of the variables and of the relationships are based on expert and prior knowledge. We precisely lack a consensual summary of all complex interactions of all social determinants of healthcare system utilization.
We added the following in the methods, covariates section: “The candidate determinants of healthcare system utilization were selected from the available data and based on existing literature, and experts’ knowledge encompassing variables related to health status, demographic characteristics, and socio-cultural and economic position.”
- Please include validation methods such as cross-validation or comparison with an established causal model for the evaluation of each structure learning. By ensuring the evaluation metrics used and the validation in place. It is important for confidence in the algorithms' outputs
Again, it is not about causal structure per se, but SL. So, there is no relevance in comparing the outputs of SL algorithms with other causal models, except the only one we could provide: the one established based on literature and experts’ knowledge. Besides, we do act on simulated data which would have been derived from a pre specified and known causal model. We act on real-world data and that’s one of the originalities and interest of our study. We comment this aspect in the discussion: “Specifically, it is difficult to validate results in real-world scenarios where the 'true' structure cannot be observed, making it impossible to measure the distance between the predicted and observed structures [8,31].”
From a technical point of view, as for the validation of the models we built, we used multiple runs on several bootstrap samples (1000 iterations, as stated in the method section) which is a robust method.
- I suggest to include a sensitivity analysis for the assessment of the model's robustness. Given the complexity of causal inference in observational data. The sensitivity analysis is important. This omission weakens the reliability of the causal conclusions drawn from the data.
As indicated in our responses to questions 3 and 4 of the first reviewer, the main goal of the study is not to assess directly the potential discovery of causal structure, but to assess the data structure in the sense of the network of complex relationships or associations between variables, may they be causal or not, and compare what can be obtained using an expert’s knowledge driven approach vs some algorithm data driven approaches. This objective is more exploratory than confirmatory. We think that the usual sensitivity analyses applied to test confounding bias, selection bias or measurement error are beyond our objective. We mainly focused on the sampling error applying a bootstrap procedure to assess the stability of our results
FOR THE RESULTS:
- There are results about network structures generated by different algorithms but it does not offer sufficient interpretative detail. For me it is unclear how readers should interpret the directionality or thickness of the links in these structures. Detailed legends as well as interpretative commentary on the link strengths can aid in clarity. Provide also comparative visualizations of network outputs across algorithms to improve network structure interpretability.
To detail the results we added the networks produced by the SL algorithms (direct access to care) in the main text.
However, we feel important to state here again that we are not searching for a causal interpretation of the graphs resulting from the application of SL algorithms on our data. We are interested in the structure, therefore in the networks of variables relationships.
Regarding how to read thickness of the links or the direction of arrows, it is specified in legends of figures, such as for figure 1:
“solid thick lines for links proposed by experts and confirmed by the non-automated epidemiological approach, thin lines for links considered as potential by experts but not confirmed by the approach. Direction of arrows was defined by expert (not data driven)”.
- The results section implies causality in some findings without sufficient backing. Causal claims should be carefully qualified, especially in observational studies, where inferring causality is challenging. Highlighting that the findings are associative rather than strictly causal would be a more accurate interpretation.
We totally agree with the reviewer. We would like to draw their attention to the fact that no causal interpretation has been made of our results – since it is not our goal.
Comments on the Quality of English Language: Moderate English revisions needed.
English has been revised.
Reviewer 3 Report
Comments and Suggestions for Authors
In this paper, the authors discuss a structure learning method within an epidemiological analysis of real-world data. The manuscript should undergo careful revision for grammatical errors and typographical issues. For example:
1. Shorter words like articles (a, an, the), coordinating conjunctions (and, or, but), and prepositions (in, of, to) are typically lowercase unless they appear as the first or last word in the title.
2. The sentence, “The specific aim was to use these networks to identify determinants of access to healthcare among various social factors” is unclear. Please clarify what is meant by "these networks".
3. The sentence “a crucial step for causal inference from observational data” does not make sense.
4. The term “Non-automated approach” is not used consistently throughout the paper.
5. The version of bnlearn and Rgraphviz packages in R should be specified.
6. The sentence “3006 people were included in the study.” is incorrect.
7. The phrase “The final network, identified by the non-automated epidemiological approach from the conceptual network proposed by experts, is given in Figure 1” is potentially misleading. Though the network was initially developed by experts, the result in Figure 1 is the product of the authors’ analysis.
8. The sentence “variables among the 11 had been identify as direct determinants” contains errors.
9. The phrase “Age, which was never retained in the structure learning approaches” is not a complete sentence. Please revise.
10. “With knowledge constraints” in line 234 appears unclear—is this intended as a subheading or was it included in error?
Comments on the Quality of English LanguageThe manuscript should undergo careful revision for grammatical errors and typographical issues.
Author Response
- Shorter words like articles (a, an, the), coordinating conjunctions (and, or, but), and prepositions (in, of, to) are typically lowercase unless they appear as the first or last word in the title.
We made changes.
- The sentence, “The specific aim was to use these networks to identify determinants of access to healthcare among various social factors” is unclear. Please clarify what is meant by "these networks".
This sentence has been replaced.
- The sentence “a crucial step for causal inference from observational data” does not make sense.
This sentence has been replaced.
- The term “Non-automated approach” is not used consistently throughout the paper.
We made changes accordingly.
- The version of bnlearn and Rgraphviz packages in R should be specified.
We added this information at the end the method section: “For the structure learning algorithms, we used the “bnlearn” package release 4.9 [28] and the Rgraphviz package release 2.46.”
- The sentence “3006 people were included in the study.” is incorrect.
This sentence has been replaced.
- The phrase “The final network, identified by the non-automated epidemiological approach from the conceptual network proposed by experts, is given in Figure 1” is potentially misleading. Though the network was initially developed by experts, the result in Figure 1 is the product of the authors’ analysis.
We clarified our method and the different aspects of the two approaches. Nonethless, we do not understand the point of the reviewer: the network is not the product of the analysis (what analysis does the reviewer refer to? Statistical or experts’ cognition?), but the result of experts’ knowledge, then tested against data to see what remains in terms of relationships given this specific dataset.
- The sentence “variables among the 11 had been identify as direct determinants” contains errors.
We corrected it.
- The phrase “Age, which was never retained in the structure learning approaches” is not a complete sentence. Please revise.
We changed the sentence.
- “With knowledge constraints” in line 234 appears unclear—is this intended as a subheading or was it included in error?
We changed the sentence.